# Improved Lycopene Production from Different Substrates by Mated Fermentation of *Blakeslea Trispora*

**DOI:** 10.3390/foods8040120

**Published:** 2019-04-09

**Authors:** Ayse Sevgili, Osman Erkmen

**Affiliations:** Department of Food Engineering, Faculty of Engineering, University of Gaziantep, 27310 Gaziantep, Turkey; aysesevgili8@gmail.com

**Keywords:** lycopene, *Blakeslea trispora*, fermentation, vegetable oil, orange powder

## Abstract

The production of lycopene from different substrates by *Blakeslea trispora* in fermentation was investigated. Lycopene productions from 4 and 6% glucose (pH 6.5) in shake flask fermentation were 77.7 and 28.1 mg L^−1^. Increasing the glucose concentration to 6% resulted in a decrease in lycopene production by 36.2%. A maximum lycopene concentration of 944.8 mg L^−1^ was detected with 4% glucose supplemented with 1.0 % sunflower oil in fermentor studies. Lycopene productions in the presence of sunflower and corn oils in the fermentor were 12.2 and 11.1 times higher, respectively, then without oil from 4 % glucose in a shake flask. Lycopene production from orange peel was two times higher in the fermentor than in the shake flask. Zygospores of *B. trispora* are the morphological forms, which are responsible for the production of the lycopene. The highest level of zygospores was correlated with the highest amount of intracellular lycopene in the total biomass dry weight. The media containing only orange powder (1%) gave a 4.9 mg L^−1^ lycopene production in a fermentor. The biosynthesis of lycopene has been started in most cases simultaneously in the early growth phase even in trace amounts. Maximum lycopene concentration was obtained when the medium was supplied with sunflower and corn oils. There is an indirect relationship between biomass and lycopene concentration.

## 1. Introduction

Lycopene is an important naturally occurring intermediate metabolite in the mevalonate metabolic pathway of microorganisms and can be transformed into β-carotene by lycopene cyclase. Thus, in order to enhance lycopene accumulation, lycopene cyclase activity should be inhibited by adding lycopene cyclase inhibitors into the fermentation medium. Lycopene has important physiological functions to prevent chronic diseases, including osteoporosis, certain types of cancer, cataract formation and cardiovascular diseases; enhance immune responses; and other biological effects in humans and animals [1,2]. In the last few years, demand on lycopene has increased due to its use in the food, pharmaceutical, cosmetic and animal feed industries [3]. Moreover, lycopene is used as an antioxidant to reduce cellular or tissue damage, and as a coloring agent for food products [4]. These make lycopene very attractive in the food and feed industries, and in medicine and cosmetic formulation. Lycopene used industrially is manufactured from plant extraction and chemical synthesis. Market needs for lycopene are not met using plant extraction due to the restricted number of plant sources and high cost. Chemical synthesis also has disadvantages including low yields, product instability, low product quality and high production cost. Due to decreased natural production of lycopene and its increased worldwide demand, there is the potential for lycopene to be produced using fermentation. Various algae and fungi produce intracellular carotenoids, including β-carotene, lycopene and astaxanthin. *Blakeslea trispora*, *Mucor circinelloides*, *Candida utilis* and *Phycomyces blakesleeanus* have the ability to produce lycopene [5]. *B. trispora* has an advantage in that it does not need any specific fermentation conditions for growth. Mating type of (+) and (−) *B. trispora* strains together can increase production of lycopene [6].

There is much interest in biotechnological production of lycopene due to consumer demand for high quality and “natural” food additives. Efforts have been made to increase the lycopene yield and reduce process costs, by using low-cost agro-food media rich in organic compounds (such as carbon, nitrogen, mineral and other sources). The amount of lycopene production depends on microorganism, the culture medium, substrates and the fermentation conditions (temperature, pH, aeration rate etc.). Most of the research has been aimed at optimizing the culture conditions that directly affect the growth of the microorganism. The main research is focused on optimization of the medium, extraction of lycopene from *B. trispora* and fermentation conditions [4,5,6,7,8,9].

An industrial mold over-producing lycopene is *B. trispora* which accumulates all-trans-carotene, as a secondary metabolite [10]. *B. trispora* is a saprophyte and performs vegetative cycle of spores and filamentous mycelia [6]. There has been a lot of effort to use food wastes in the production of lycopene mitigation [11]. At present, there are no research on the use of orange powder and sunflower oil in the production of lycopene. The transport cost, sales problems, low quality and susceptibility to spoilage of the food-wastes by microorganisms have led to alternative utilization approaches. Agro-food wastes (fruits and vegetables: cabbage residues, watermelon husk and peach powders) have been used in the production of carotenes by *B. trispora* [12,13].

Optimization of fermentation conditions can also be required in the production of lycopene from different nutrient sources. This report was focused on the effect of a series of media containing various substrates (glucose, refined vegetable oils and orange powder) in the accumulation of lycopene in submerged shake flasks and fermentor studies. Morphological, biomass and pH changes occurring during fermentations were also studied.

## 2. Materials and Methods

### 2.1. Microorganism and Chemicals

*Blakeslea trispora* mating strains ATCC 14,271 (+) and ATCC 14,272 (−) were obtained from the American Type Culture Collection (ATCC; Rockville, MA, USA). Lyophilized cultures were rehydrated with malt extract broth (MEB; Darmstadt, Germany) by incubation at 35 °C for 3 days and were used in the stock culture preparation. Stock cultures of *B. trispora* were prepared on potato dextrose agar (PDA; Darmstadt, Germany) slant. Yeast extract, glucose, agar, asparagine, starch, lycopene, nicotine, KHPO_4_, MgSO_4_, HPLC grade petroleum ether, methanol, acetonitrile and dichloromethane were obtained from Sigma-Aldrich (Interlab Company, Adana, Turkey).

### 2.2. Preparation of B. Trispora Cultures

*B. trispora* (+) and (−) cultures were prepared for fermentation studies using a yeast phosphate soluble starch (YpSS) agar medium. The composition of YpSS agar medium (g L^−1^) was: Yeast extract 4.0, K_2_HPO_4_ 1.0, MgSO_4_ 0.5, starch 15.0 and agar 20.0. The pH of the medium was adjusted to 6.5. Each of the *B. trispora* strains from stock culture was inoculated separately onto YpSS agar plates and incubated at 28 °C for 5 days. After sporulation of cultures, 10 mL of sterile distilled water was added onto the cultures in the Petri plates and the spores were collected by scraping off the medium surface using a sterile cotton swab. The spore suspension containing 3.9–8.5 × 10^6^ spores ml^−1^ was used to inoculate the fermentation media. Strains were inoculated into the fermentation medium with 2.5 and 7.5 mL of spore suspension (+) and (−) strains, respectively, to provide 1:3 ratio of inoculation.

### 2.3. Preparation of Fermentation Media

The basal fermentation medium (BFM) had the following composition (g L^−1^): Yeast extract 1.0, asparagine 2.0, K_2_HPO_4_ 1.5 and MgSO_4_ 0.5. BFM was supplemented with different concentrations of ingredients as given in Table 1: Refined vegetable oils (sunflower and corn oils), glucose and milled orange peel (powder). Samples of orange (*Citrus sinensis*) peel were dried at 40 °C to reach a final moisture lower than 10%, milled to a particle size (250 µm) using a laboratory grinder (Roller type lab mill RM 1300, Erkaya, Ataşehir, İstanbul, Turkey), homogenized in a single lot to avoid any variation in composition, and stored at 4 °C in a cold chamber until use. Fermentation media were prepared at two different pH: 6.5 and 5.5. Fermentation media (50 mL) in Erlenmeyer flasks (250 mL) were sterilized at 121 °C for 15 min in autoclave. Fermentation media for fermentor studies were sterilized in a 7 L Bioflow 410 sterilizable-in-place benchtop fermentor (New Brunswick Scientific BioFlo 410 SIP Fermentor, Eppendorf, New York, MA, USA).

### 2.4. Shake Flask Fermentation Process

Fermentation media (50 mL) in Erlenmeyer flasks (250 mL) were inoculated with 10 mL *B. trispora* strains ATCC 14,271 (+) and 14,272 (−) by 1:3 ratio from spore stock culture. The lycopene cyclase inhibitor of nicotine (5 mM) was added on the second day of fermentation. Fermentation media (total volume 60 mL) were incubated at 28 °C for 7 days on a bench type water bath shaker ST-402 (NÜVE; Sanayi Malzemeleri Imalat ve Ticaret A. Ş., İstanbul, Turkey) with wrist shaking at 120 rpm.

### 2.5. Fermentation Process in Fermentor

All fermentations were performed in the Bioflow 410 fermentor with a working volume of 5.0 L. Sterilization sequences in this fermentor are fully automatic and heat-up and cool down in under 30 min. All easily initiated using the advanced touch-screen controller. The fermenter was equipped with a pH probe, dissolved oxygen probes, automatic sampler, temperature probe, inoculation port, sparger, exhaust condenser, agitation, vessel light, level sensor and BioCommand SCADA Software (RS-232; New Brunswick Scientific BioFlow 410 SIP Fermentor, Eppendorf, New York, MA, USA).

Four liter of fermentation media (from BFM) were prepared in the fermentor. The initial pH of fermentation was adjusted to 6.5. The fermentation with 4% glucose media was also performed at two aerations and agitations. The fermentor was incubated at 28 °C. The inoculation ratio of two strains (+/−) into the fermentation medium from plate cultures was the same as those in the shake flask. The lycopene cyclase inhibitor of nicotine (5 mM) was added on the second day of fermentation. Dissolved oxygen tension was not controlled during fermentation.

### 2.6. Sampling

Fermented samples were aseptically removed from the shake flask and fermentor through automatic sampler after 2, 3, 4, 6 and 7 days of fermentation. Initial samples were also removed at the beginning of fermentation. At every sampling interval, 5 mL of sample was taken from the fermentation media under aseptic conditions and the contents were analyzed for lycopene, pH and biomass changes. Two ml of the sample was used in lycopene analysis and 3 mL was used in pH and biomass analysis. At each sampling time and after the first day, 0.4 mL of the sample was also removed for morphological analysis.

Two samples were removed and two parallel analyses were performed from each sample. All experiments were repeated three times.

### 2.7. Analysis

#### 2.7.1. Lycopene Analysis

Samples were analyzed for lycopene by a High-Performance Liquid Chromatography (HPLC) method [1]. Two ml of a 5 mL sample was mixed with 15 mL of petroleum ether. The mixture was subjected to ultrasound (Soniprep 150 Ultrasonic Disintegrator, MSE, London, UK) for 30 s and then, the sample was centrifuged using a table-type centrifuge (Hettich eba III) at 6000 rpm for 15 min. The supernatant was passed through 4 µm filter paper. About 2.5 mL of liquid was added into a 3 mL vial and the sample was analyzed by HPLC immediately. The presence of lycopene was detected by HPLC using a fluorescence detector. The flow rate was 1 mL min^−1^ and the column was nukleosil C18 (250 × 4.6 mm ID). Temperature was 28 °C. Lycopene was detected at 450 nm and 20 µL sample was injected to HPLC automatically. The result was read for 40 min. The mobile phase was acetonitrile-methanol-water-dichloromethane (7:1.5:0.5:1, v:v:v:v). The mobile phase was filtered through a disposable filter unit (0.45 μm) and degassed in a degasser (Bransonic 2200, 41 Eagle Road Danbury, Connecticut, 06810-1961, CT, USA). The peaks in a chromatogram were evaluated according to the lycopene standard curve.

*Analytical Method Validation:* Linearity was determined between 0.1 to 0.0001 g ml^−1^ using five levels of calibration in triplicate. The Mandel’s fitting test was used to evaluate the linearity of the straight-line regression model [14].

Limit of detection (LOD) and limit of quantification (LOQ) were determined according to the method described by Cucu et al. [15]. For this, standards of 0.1 to 0.0001 g lycopene ml^−1^ were prepared and injected three times each. The mean of the slopes (S) and standard deviation of the intercepts (σ) were calculated from the three calibration curves. The LOD and the LOQ were calculated according to the formulas below [15]:LOD = (3.3∗σ)/SLOQ = (10∗σ)/S

For the assessment of the matrix effect, calibration curves were prepared in the petroleum ether and analyzed in triplicates. The calibration curve of lycopene was the equations: y = ((area+intercept)/slope).

#### 2.7.2. pH, Biomass and Morphological Analysis

In biomass analysis, Whatman No. 41 filter paper was dried at 105 °C to constant weight, cooled in a desiccator and weighed before use. Three ml of the remaining sample was filtered through the filter paper. The filter cake on paper was washed three times with distilled water. The filter paper was placed into an incubator at 105 °C, dried until constant weight and then weighed to calculate the biomass (g L^−1^). The medium with %10 orange powder was used as a control; in biomass detection, this was subtracted from biomass.

In pH analysis, a filtered solution was filled to 10 mL with distilled water and pH was detected using a pH meter (EMAF EM78X model, Interlab Company, Adana, Turkey) equipped with a glass electrode.

In image analysis, the morphological changes of *B. trispora* during the fermentation was analyzed using an image analysis system consisting of a light microscope (Olympus BX51; Olympus Corporation, Shinjuku Monolith, 3-1, Nishi Shinjuku 2-chome, Shinjuku-ku, Tokyo, Japan) equipped with a Pixera PVC 100C camera. Samples for image analysis were prepared by mixing 0.4 mL of fermentation broth with 0.1 mL of lactophenol blue. One drop of the mixed sample was pipetted on to a microscope slide, covered by a cover slip so that no air gap or bubbles were trapped between the sample and the coverslip and observed immediately with 20X objective by taking 5 different images. The morphological parameters measured were the area of zygospores and zygospores (% of total area of mycelium), the diameter of zygospores (mm), freely dispersed hyphae and clumps (aggregates) of different sizes on the basis of their projected area, and the hyphae length (mm). The microscope eyepiece is equipped with a scale micrometer that is used to measure the morphological parameters of mold. For each sample, the process was repeated at least 5 times using new positions on the same slide, and the morphological parameters were expressed as the mean values of each sample [16].

### 2.8. Statistical Analysis

All data were analyzed by SPSS 16 software (SPSS Inc., Chicago, IL, USA). Statistical differences between fermentation time (days) and between substrates used in the fermentations were tested by one-way analysis of variance following Duncan’s multiple-range test. A probability of 0.05 was used to determine statistical significance.

## 3. Results

The fermentations were carried out at 28 °C for 7 days using different carbon sources and ingredients in the shake flask and fermentor.

The initial pH is one of the important parameters in the production of lycopene by *B. trispora*. The pH of the fermentation media changed during the fermentation time. The pH of the fermentation media was decreased during the first 2 days then increased during the remaining 7 days. At 2 days, the pH reduced to approximately 5.0, and then the pH increased over 6.0. The pH drop would be due to trisporic acid or/and other unidentified acidic formations in the lycopene production [17]. According to Nanou et al. [18], the pH of the media decreased slightly during the first 2 days of the fermentation from 7.6 to 5.3 and then increased slowly up to 7.0 at the end of the fermentation. The pH of the fermentation media was probably increased by ammonia liberation during degradation of proteins in the media by *B. trispora*. This would be indicative of amine metabolic product being released as a consequence of cell death and subsequent lysis occurring after that time.

### 3.1. Shake Flask Studies

Lycopene production of two glucose concentrations (4 and 6%) were investigated at two initial pH values (6.5 and 5.5). Figure 1 and Figure 2 show the effect of glucose concentrations on lycopene production at the initial pH 6.5 and 5.5 respectively. Lycopene production at pH 6.5 with both glucose concentrations was higher than at pH 5.5. The amount of lycopene production was 77.7 and 28.1 mg L^−1^ with 4 and 6% glucose, respectively, after 7 days fermentation at pH 6.5. The biomass dry weight (DW) increased with the increase in glucose concentration from 4 to 6%. The lowest amount of lycopene content (0.3 mg g^−1^ DW) was obtained at pH 5.5 with glucose concentration 6%. In this research, 4% glucose yielded better lycopene than 6% while the biomass is better produced in 6% glucose. Since, there is an indirect relationship between biomass and lycopene concentration. Subsequently, increasing the glucose concentration to 6% resulted in a decrease in lycopene production by 36.2%. The decreased concentration of lycopene encountered with the highest concentration of glucose was probably due to osmotic effects. It has been reported that above a critical substrate concentration, the decreased water activity and the onset of plasmolysis combine to cause a decrease in the rates of fermentation and product formation [19]. In all cases, the biomass DW increased during the first two days of the fermentation and then remained constant until the maximum concentration of the lycopene was obtained. The maximum lycopene content was 2.4 and 0.6 mg g^−1^ DW with 4 and 6% glucose, respectively, at pH 6.5 after 7 days while they were 0.7 and 0.1 mg g^−1^ DW after 4 days, respectively. There are significant differences (*p* < 0.05) in lycopene production during fermentation. There are also significant differences (*p* < 0.05) in lycopene production between 4 and 6% glucose concentrations, and pH 6.5 and 5.5 at both glucose concentrations.

Figure 3 and Figure 4 show the effect of sunflower and corn oils, respectively, on the production of lycopene in 4 % glucose medium at initial pH 6.5. The maximum lycopene was produced 795.1 and 714.2 mg L^−1^ with 1 % sunflower and corn oils respectively. The highest amount of lycopene (335.4 mg L^−1^) was produced with 3 % sunflower oil rather 3 % corn oil (275.3 mg L^−1^). Lycopene content was 131.4 and 115.0 mg g^−1^ DW at the end of fermentation for 1% sunflower and corn oils respectively. There are significant differences (*p* < 0.05) in lycopene production during the 7 days of fermentation. There are also significant differences (*p* < 0.05) in lycopene production with and without oil, and between both oil types.

Lycopene productions with 1 and 2% orange powder during 7 days of shake flask fermentation were given in Figure 5. Lycopene productions after 7 days were 2.9 and 1.7 mg L^−1^ with 1 and 2% orange powder respectively. Biomasses formations were 8.6 and 11.6 g L^−1^ for orange powder respectively. Lycopene content was 0.3 and 0.1 mg g^−1^ DW respectively at the end of fermentation. There were significant differences (*p* < 0.05) in lycopene production during the 7 days of fermentation for 1 and 2% orange powder, between the two orange powder concentrations.

### 3.2. Fermentor Studies

Effects of 4 and 6% glucose concentration (pH 6.5) on the production of lycopene were studied at 3 L min^−1^ aeration with 500 rpm agitation (Figure 6). The highest amount of lycopene was produced (92.3 mg L^−1^) with 4% glucose concentration rather than 6% glucose (57.8 mg L^−1^). Biomass DW formations after 7 days of fermentation were 154.9 and 196.7 mg L^−1^ with 4 and 6% glucose concentration respectively. Lycopene content was 3.8 and 1.9 mg g^−1^ DW at the end of fermentation for 4 and 6% glucose concentrations respectively. There is an indirect relationship between biomass and lycopene formation similar to the shake flask. The production of lycopene in the fermentor was higher than in the shake-flask from glucose. This would be due to the better aeration of the medium in the fermentor. There were significant differences (*p* < 0.05) in lycopene production during the 7 days of fermentation. There were also significant differences (*p* < 0.05) in lycopene production between 4 and 6% glucose concentrations.

The effect of high level of aeration (12 L min^−1^) and agitation (750 rpm) on the production of lycopene in a 4% glucose medium during the 7 days of fermentation is given in Figure 7. The highest amount of lycopene was produced at low level of aeration (3 L min^−1^) and agitation (500 rpm) rather than at the high levels (12 L min^−1^ and 750 rpm). The highest amount of biomass was formed at the higher aeration (12 L min^−1^) and agitation (750 rpm) rather than at the lower levels. Lycopene content of DW (3.8 mg L^−1^) was higher at 3 L min^−1^ aeration and 500 rpm agitation rather than 12 L min^−1^ at 750 rpm (1.8 mg L^−1^). Generally, the results showed that the biomass increased with increasing aeration and agitation. This was due to the better air supply to the cells. This is especially important for high biomass concentrations. *B. trispora* is a strictly aerobic microorganism. Thus, moderate aeration rates in combination with high impeller agitation improved the growth of mold. There are significant differences (*p* < 0.05) in lycopene production during the 7 days of fermentation and between the two fermentation conditions (3 and 12 L min^−1^ aeration).

Figure 8 shows the effect of 1% sunflower and corn oils on the lycopene production in the fermentor. Lycopene production was 944.8 and 859.8 mg L^−1^ respectively. Biomass formations were 29.5 and 19.6 g L^−1^. Lycopene content was 65.0 and 48.9 mg g^−1^ DW. Lycopene production in the presence of sunflower and corn oils in the fermentor was 12.2 and 11.1 times higher, respectively, then without oil from 4% glucose in a shake flask. This would be due to retaining higher amount of oxygen in the media. Since oil slightly increases the density of the fermentation media. There are significant differences (*p* < 0.05) in lycopene production during fermentation periods. There are also significant differences (*p* < 0.05) in lycopene production with and without oil, and between sunflower and corn oils.

Lycopene production was studied from different concentrations of orange peel in 4 L fermentation media with aeration 3 L min^−1^ and agitation 500 rpm (Figure 9). Lycopene was produced 4.9 mg L^−1^ with 1% orange peel. Biomass was 9.7 g L^−1^ and lycopene content was 0.05 mg g^−1^ DW at the end of fermentation. Lycopene production from orange peel was two times higher in the fermentor than in the shake flask. There are significant (*p* < 0.05) differences in lycopene production during the 7 days of fermentation.

### 3.3. Morphological Analysis

The morphology of *B. trispora* was studied in the shake-flask and fermentor studies (3 L min^−1^ aeration and 500 rpm agitation) in only 4% glucose medium containing 1% sunflower oil using an image analysis system. During the first day of fermentation, microscopic examination showed that *B. trispora* formed a great amount of mycelium in the fermentation broth. After one day of fermentation, two opposite mating types of *B. trispora* produced zygospores which grew towards each other and produced pro-gametangia at their tips. Septation in the pro-gametangia led to the production of terminal gametangia, which fused to form the zygospores. The area of zygospores to the total area of mycelium remained almost constant up to 4 days of fermentation and then decreased significantly (*p* < 0.05) and disappeared on the 7th days of fermentation. On the other hand, the area of zygospores to the total area of mycelium increased significantly (*p* < 0.05) with the increase of fermentation time from 4 to 7 days. The hyphae length increased from the first day of fermentation up to 4 days and then slightly increased until the end of the fermentation. The diameter of zygospores increased from the first 2 days of incubation up to 4 days and then decreased slightly. Generally, morphological measurements using image analysis showed that *B. trispora* formed zygophores, zygospores and mycelium in which the hyphae appeared as a homogeneous dispersed suspension through the fermentation medium.

## 4. Discussion

Various concentration of carbon sources and refined natural vegetable oils as co-substrates were used in the production of lycopene. The concentration of carbon sources and vegetable oils greatly affected the final amount of lycopene. Furthermore, the use of orange powder components gives a first indication that *B. trispora* is able to metabolize and produce lycopene.

The highest concentration of lycopene was produced from a low concentration of glucose (4%) rather than at the high concentration (6%) in both the shake flask and fermentor studies. High concentrations of glucose may cause inhibition of the activity of mevalonate kinase [20], which is one of the key enzymes of the lycopene biosynthesis pathway. The carbon source can act as a major constituent for the building of cellular material and as an important energy source during the microbial fermentations [21]. Lycopene production in all fermentation conditions was lower than 1.0 mg L^−1^ during the first 2 days of fermentation and continuously increased during the next 5 days. It reached maximum concentration after 7 days. Sugar metabolism was slightly enhanced at the beginning of fermentation and increased during next fermentation period. This increase may provide more lycopene production [22]. On the other hand, biomass formation increased continuously during the 6 days of fermentation and it slowed on last day. When lycopene production increased, biomass formation decreased. Dry weight increased continuously up to 6 days and then slowed down. But lycopene production rate was higher on the last day. The maximum lycopene content in cells occurred after 7 days of fermentation when compared with 6 days in all fermentation conditions. This indicates that lycopene and biomass formations are inversely proportional. The biomass rapidly increased after 2 days of fermentation and this continued up to 6 days of fermentation. Nanou et al. [18] reported that the concentration of residual sugars fell rapidly during the first 4 days of the fermentation after which it decreased slowly and this was accompanied by a rapid increase of biomass concentration.

The refined vegetable oils stimulate different biosynthetic pathways in *B. trispora*. In this study, about 944.7 mg L^−1^ of lycopene was produced in fermentor studies from 4% glucose supplemented with 1 % sunflower oil. With 1% corn oil, about 859.8 mg L^−1^ of lycopene was produced in a 4% glucose medium. These results indicate that sunflower oil has a greater influence on lycopene production than corn oil due to a higher amount of vitamin E and linoleic acid in sunflower oil [23,24]. Similar results were demonstrated by Mantzouridou et al. [7] that corn steep liquor, olive oil, soybean oil, cottonseed oil and linoleic acids significantly increased the β-carotene production. Mantzouridou et al. [24] was also indicated that the addition of 10 g oil L^−1^ of substrate stimulated β-carotene production. Beside triacylglycerol, the main component of oils can be used in the lipid biosynthesis (involving acetyl-CoA formation resulting from β-oxidation of fatty acids) of the mold mycelium. Oils can be hydrolyzed to fatty acids and glycerol by mold exolipases [23]. When the concentration of natural oils increases (from 1 to 3%), the production of the lycopene decreases while the biomass dry weight increases significantly. This may be related to the consumed amount of the oils converted to biomass instead of lycopene. The production of lycopene without oils in the medium was lower while biomass formation was higher.

Two different aerations (3 and 12 L min^−1^) and agitations (500 and 750 rpm) were studied in a 7 L fermentor (pH 6.5). The highest amount of lycopene 944.8 mg L^−1^ was produced with 3 L min^−1^ aeration and 500 rpm agitation in 4% glucose medium containing 1% sunflower oil. The positive impact of oils in the production of lycopene has been confirmed in other reports, such as those based on the use of crude olive or soybean olive [7,24] or waste cooked oil [25]. Supplementation of the medium with industrial glycerol, obtained either from soap manufacturing or biodiesel production industries, allows up to a tenfold increase in β-carotene levels [26]. Very promising results have been also obtained using agro-food wastes rich in carbohydrates and mineral salts, as beet molasses [27], cheese whey [28]. The concentration of lycopene was higher in fermentor studies than in the shake flask. Aeration could be beneficial to the growth and performance of aerobic *B. trispora*. Agitation of the fermentation medium creates shear forces which affect microorganisms in several ways: causing morphological changes, variation in their growth and product formation, and damaging the cell structure [19].

Orange powder was used as a raw material for production of lycopene in a shake flask and fermentor. The highest amount of lycopene (4.8 mg L^−1^) was produced in the fermentor with 1% orange powder rather than in the shake flask (2.9 mg L^−1^). Citrus peels are rich in cellulose, hemicelluloses, proteins and pectin, but the fat content is low. Pourbafrani et al. [21] reported that more than 95% of the peel oil is D-limonene which is extremely toxic to fermenting microorganisms. The addition of tomato sauce as a waste into the fermentation medium enhanced the production of lycopene and its production was 3.52 mg/100 g cell DW [29].

*B. trispora* is a micro-fungus with a life-cycle involving hyphae, zygophores and zygospores in submerged fermentation. The zygophores are the precursors of zygospores, which are responsible for the production of lycopene. [19]. The highest percentage of zygospores in an observed microscope field was correlated with the highest percentage of intracellular lycopene in the total biomass dry weight after the 7 days of fermentation. High percentages of vacuolated hyphae, evacuated cells and degenerated hyphae of *B. trispora* were observed at higher aeration (12 L min^−1^). The percentage of the vacuolated hyphae, evacuated cells and the degenerated hyphae formed during the first 2 days of fermentation were higher than the remaining fermentation time. The growing mycelium was composed primarily of intact hyphae (such as 96.8% of the biomass dry weight), and zygospores on the 2nd day of fermentation, whereas on the next day the concentration of the intact hyphae was reduced to 67.9 of the biomass dry weight and the concentration of the zygospores was about 8.3%. On the 7th day of fermentation, the mycelium was composed mainly of evacuated cells. The addition of 1% oils to the fermentation medium increased the concentration of the intracellular lycopene during fermentation and reached to maximum level after 7 days, whereas the percentage of zygospores to the biomass dry weight increased significantly (*p* < 0.05). The highest concentration of lycopene correlated with the highest percentage of zygospores in the biomass DW. When the natural oils were used in the medium with 4% glucose in the fermentor, a great increase in the amount of lycopene occurred after the days 7 of fermentation, while the concentration of evacuated cells and the number of zygospores increased significantly (*p* < 0.05).

The effect of aeration rate and agitation speed on culture morphology was also observed. At an aeration rate of 3 L min^−1^ with 500 rpm agitation, the area of zygospores increased significantly and then remained constant. On the other hand, at aeration rates of 12 L min^−1^ with 750 rpm agitation, the area of zygospores decreased and then remained constant. The maximum concentration of lycopene was observed in the same conditions where a large number of zygospores were observed. This means that the zygospores were responsible for the synthesis of lycopene.

## 5. Conclusions

In this study, various substrates are compared with lycopene production in the shake flask and fermentor. The amount of lycopene production is based on the concentration of carbon sources and natural oil content. The amount of orange powder used in the fermentation as carbon sources affected the formation of lycopene. The use of natural oil is important for stimulation of the biosynthetic pathway in *B. trispora.* The medium with natural oil led to more lycopene than only carbon source in the medium. The oils degraded into acetyl-CoA, which is the precursor of lycopene derived from mevalonic acid. The mated fermentation process for lycopene production may be useful and a reference to the other fermentation. Also, lycopene production from *B. trispora* is better than other alternative plant extracts which is one of the other methods of lycopene production. For the future, this study is provided to contribute to the development in the lycopene production industry from this method. Oils improved lycopene production and enhanced vacuolation and evacuation of the hyphae. The morphological characteristics of the hyphae and the proportion of zygospores to the biomass DW were changed significantly (*p* < 0.05) within the medium composition and fermentation conditions. Moreover, there was a parallel relationship between the area of zygospores and production of lycopene. Generally, the results showed that *B. trispora* is a micro-mold that produces lycopene with a life-cycle involving hyphae, zygophores and zygospores. The hyphae do not play a role in the biosynthesis of lycopene. The zygphpores are the precursors of zygospores, which are responsible for the production of the lycopene. Maximum lycopene concentration was obtained when the medium was supplied with sunflower and corn oils.

## Figures and Tables

**Figure 1 foods-08-00120-f001:**
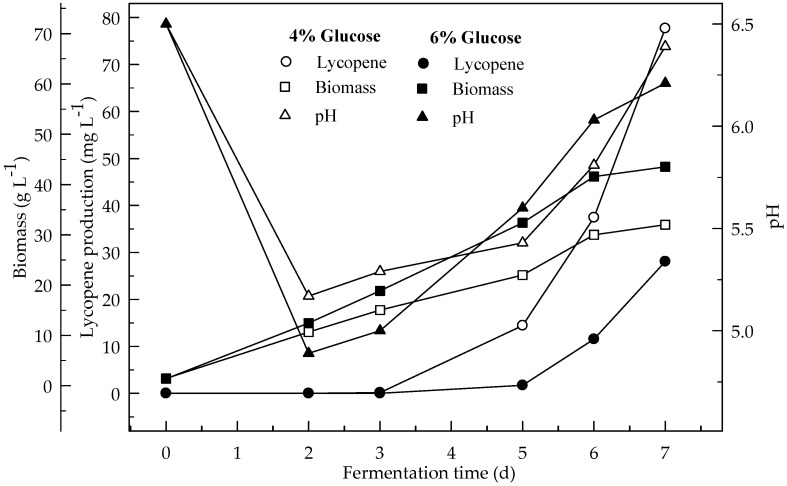
Changes in lycopene production, biomass formation and pH with initial pH 6.5 in a medium containing 4 and 6% glucose.

**Figure 2 foods-08-00120-f002:**
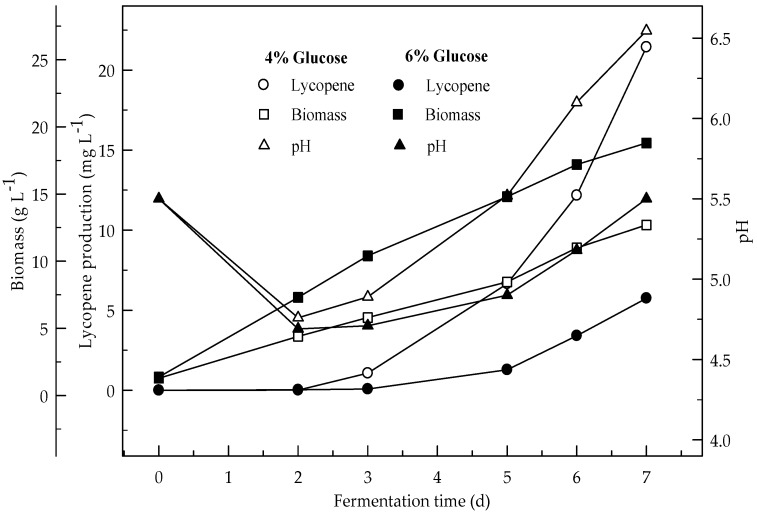
Changes in lycopene production, biomass formation and pH with initial pH 5.5 in a medium containing 4 and 6% glucose.

**Figure 3 foods-08-00120-f003:**
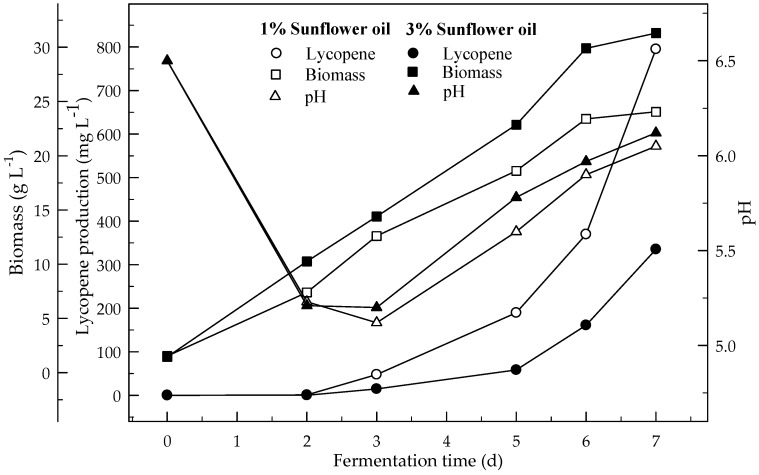
Changes in lycopene production, biomass formation and pH in fermentation with an initial pH 6.5 in a 4% glucose medium containing 1 and 3% sunflower oil.

**Figure 4 foods-08-00120-f004:**
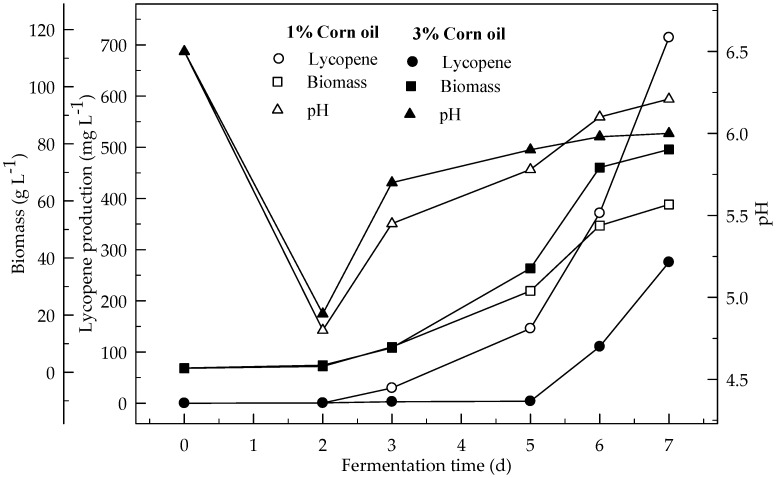
Changes in lycopene production, biomass formation and pH in fermentation with an initial pH 6.5 in a 4% glucose medium containing 1 and 3% corn oil.

**Figure 5 foods-08-00120-f005:**
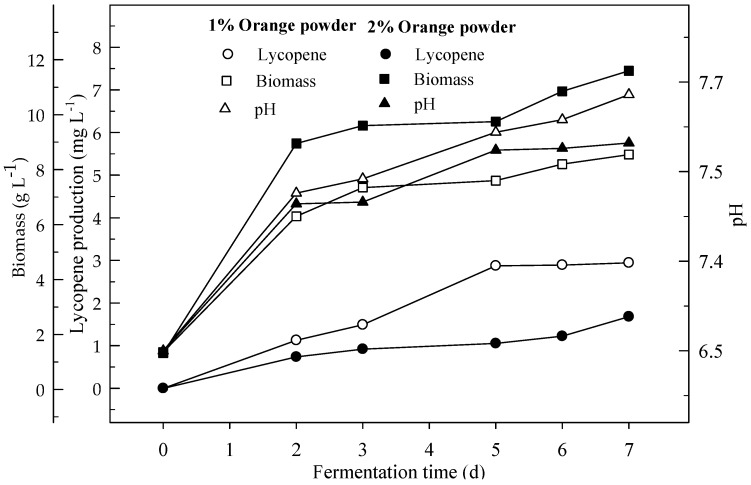
Changes in lycopene production, biomass formation and pH in a shake-flask fermentation with an initial pH 6.5 in a medium containing 1 and 2% orange powder.

**Figure 6 foods-08-00120-f006:**
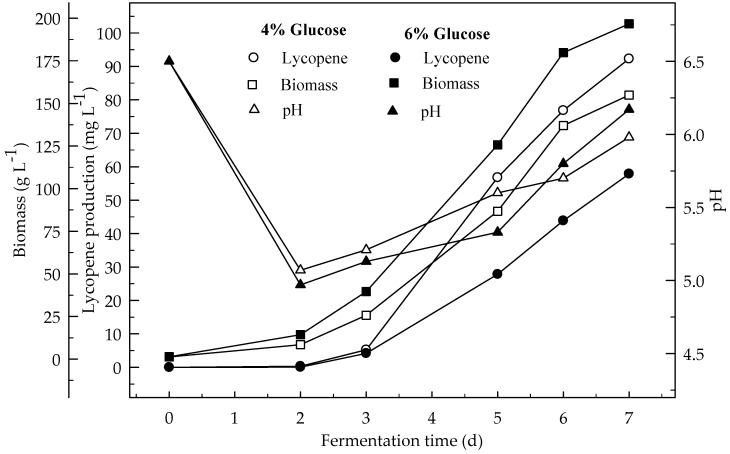
Changes in lycopene production, biomass formation and pH in a fermentor with an initial pH 6.5 in a 4 and 6% glucose medium at 3 L min^−1^ aeration and 500 rpm agitation.

**Figure 7 foods-08-00120-f007:**
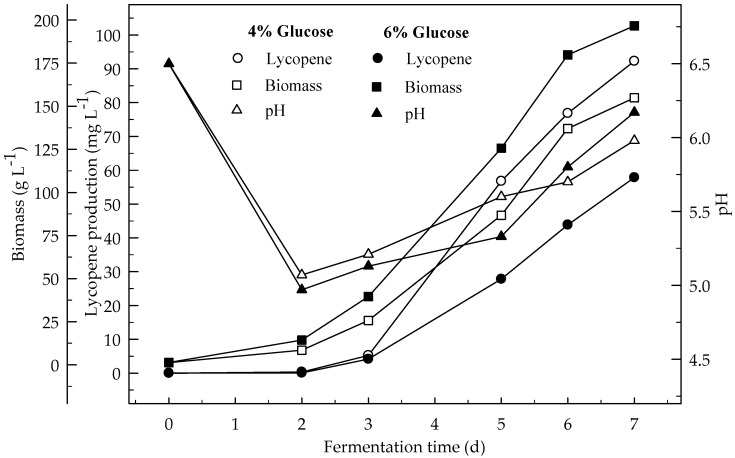
Changes in lycopene production, biomass formation and pH in a fermentor with an initial pH 6.5 in a 4 and 6% glucose medium at 12 L min^−1^ aeration and 750 rpm agitation.

**Figure 8 foods-08-00120-f008:**
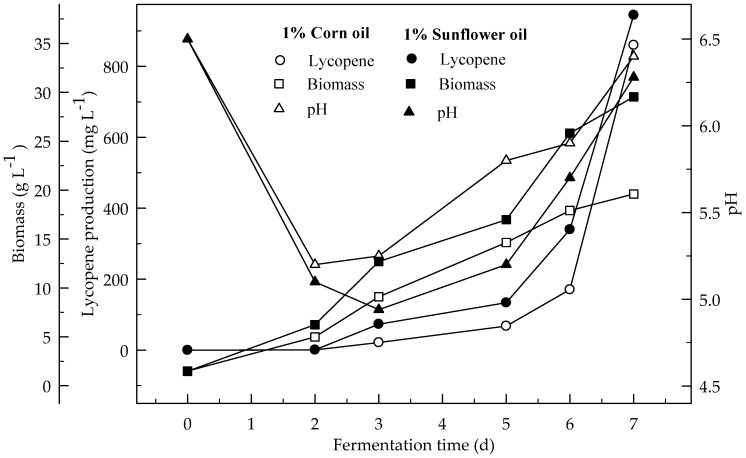
Changes in lycopene production, biomass formation and pH in a fermentor with an initial pH 6.5 in a 4% glucose fermentation medium 1% corn and sunflower oils (at aeration 3 L min^−1^ with 500 rpm agitation).

**Figure 9 foods-08-00120-f009:**
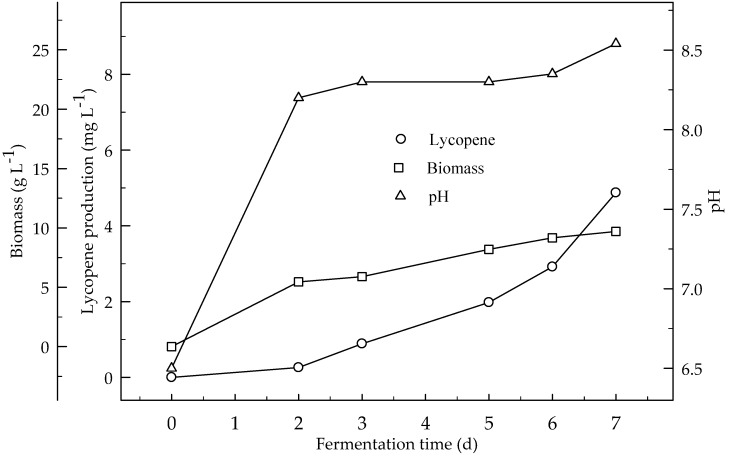
Changes in lycopene production, biomass formation and pH in a fermentor with an initial pH 6.5 from a 1% orange powder in a fermentor with aeration 3 L min^−1^ at agitation 500 rpm.

**Table 1 foods-08-00120-t001:** Medium composition.

Medium Number	Composition			
	Glucose (g L^−1^)	Sunflower Oil (ml L^−1^)	Corn Oil (ml L^−1^)	Orange Powder (g L^−1^)
**In shake flask studies**				
1	40	-	-	-
2	60	-	-	-
3	40	10	-	-
4	40	30	-	-
6	40	-	10	-
7	40	-	30	-
8	-	-	-	10
9	-	-	-	20
**In fermentor studies**				
10	40 ^a,b^	-	-	-
11	60 ^a^	-	-	-
12	40 ^a^	10	-	-
13	40 ^a^	-	10	-
14	-	-	-	10 ^a^

^a^ At 3 L min^−1^ aeration and 500 rpm agitation. ^b^ At 12 L min^−1^ aeration and 750 rpm agitation.

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
