# Peer review of "Improved Lycopene Production from Different Substrates by Mated Fermentation of Blakeslea Trispora"

_foods, 2019, doi:10.3390/foods8040120_

Round 1
Reviewer 1 Report
The manuscript present good results concerning lycopene production by Blakeslea trispora. However, the manuscript needs significant improvement in some points before any consideration for publication.
The manuscript entitled “Improved lycopene production from different substrates by mated fermentation of Blakeslea trispora” investigates lycopene production through fermentation of mated Blakeslea trispora mould. The authors utilized several substrates including renewable resources, such as orange peels. It is worth noting that provide new insights for the higher production of lycopene using oils. This give the perspective of crude and waste oils utilization. This perspective it will be good to presented in the conclusion section. Generally, the manuscript is well structured. It is recommended to be checked by a native English speaker, since there are some grammatical mistakes. Below you may find some corrections/suggestions for further improvement of the manuscript:
Abstract: it is strongly recommended to enrich the abstract with one more sentence at the end highlighting the significance of the present study.
lines 60-63: please make a separate last paragraph in the introduction section stating the objectives of the study.
line 83, section 2.3: a short paragraph should be included concerning the preparation of media no. 8 and 9 with orange peels. What was the procedure followed? Orange peels were used as solid particles in the basal medium? Were orange peels dried, grinded (particle size?)? Please give details.
section 2.7, Analysis: Monitoring of carbon source could give results for better understanding of metabolism, so why the authors did not analyze the sugar consumption during fermentation? For instance, if you had the profile of sugars you could provide better discussion for the statement “Lycopene production continuously increased during the next 5 days. It reached maximum concentration after 7 days.” and see if there is any correlation between sugars consumption and lycopene production.
Morphology was studied only in 4% glucose medium containing 1 % sunflower oil (as you mention in line 287)? Please make it clear in the materials and methods section. How biomass was monitored in the case of fermentations with orange peels? Usually fungus attach in solid particles. Clarify this with a few sentences.
line 117: please write “High Performance Liquid Chromatography (HPLC)” and thereafter use only the abbreviation.
line 116, section 2.7.1: Was the sample automatically injected? The volume of injected sample was 20μm or 20 μl?
lines 257-265: the paragraph must rewritten. There are small sentences without continuity, for example “Since oil slightly increases the 262 density of the fermentation media.”
line 261: correct “then” to “than”
section 3.3 Morphological analysis: Do you have any images from morphological analysis to provide? It would add value to the manuscript. Explain why you present results from only one media.
Discussion section: What is the content of total phenolic content in lycopene? Do the authors validate in some way if there are significant quantities of other carotenoids? What is the yield of carotenoids in terms of the biomass?
line 308: change “(6 %)” to “(6%)”
line 309: delete “the” from the sentence “cause the inhibition”
line 329: give reference for the statement “…due to a higher amount of vitamin E and linoleic acid in sunflower oil.”
lines 324-335: enrich your discussion with other publications. For example similar study with utilization of oils in the substrate has been demonstrated by Mantzouridou et al. 2006. Performance of Crude Olive Pomace Oil and Soybean Oil during Carotenoid Production by Blakeslea trispora in Submerged Fermentation. J. Agric. Food Chem., 2006, 54 (7), pp 2575–2581
lines 344-350: enrich your discussion with other publications using renewable resources as substrates.
line 357: change “12 V/min” to “12 V min-1”.
Figures: error bars are missing from the figures. Please add them. Also, correct “Lycopen” to Lycopene” in the axis title and throughout the manuscript.
Figure 1: please check the format of x-axis title for lycopene production, it should be “mg L-1”
Figure 8: change “Sunflavor oil” to “Sunflower oil” in the figure legend. Also, correct the figure caption since you mention that the medium contain both corn and sunflower oils. Also, be constant with units, for instance “3 L min-1” instead of “3 L/min”. Check for such mistakes all figure captions and the manuscript.
Author Response
The manuscript have revised according to the comments of 2 reviewers and corrections indicated under each comments of reviewers.
Other revisions on the bases of reviewer indication also indicated at the end of this file.
Comments and Suggestions for Authors
The manuscript present good results concerning lycopene production by Blakeslea trispora. However, the manuscript needs significant improvement in some points before any consideration for publication.
The manuscript entitled “Improved lycopene production from different substrates by mated fermentation of Blakeslea trispora” investigates lycopene production through fermentation of mated Blakeslea trispora mould. The authors utilized several substrates including renewable resources, such as orange peels. It is worth noting that provide new insights for the higher production of lycopene using oils. This give the perspective of crude and waste oils utilization. This perspective it will be good to presented in the conclusion section. Generally, the manuscript is well structured. It is recommended to be checked by a native English speaker, since there are some grammatical mistakes. Below you may find some corrections/suggestions for further improvement of the manuscript:
Abstract: it is strongly recommended to enrich the abstract with one more sentence at the end highlighting the significance of the present study.
The abstract have been revised and entiched with more than one sentences. Repeating sentences have been removed from manuscript:
Added sentence: “Increasing the glucose concentration to 6% resulted in a decrease in lycopene production by 36.2%. Lycopene productions in the presence of sunflower and corn oils in the fermentor were 12.2 and 11.1 times higher, respectively, then without oil from 4 % glucose in a shake flask. Lycopene production from orange peel was two times higher in the fermentor than in the shake flask. The biosynthesis of lycopene has been started in most cases simultaneously in the early growth phase even in trace amounts. Maximum lycopene concentration was obtained when the medium was supplied with sunflower and corn oils. There is an indirect relationship between biomass and lycopene concentration.”
Removed sentences from abstract to prevent repeating according to new added sentences: “The presence of oils as substrates resulted in enhanced mold growth and subsequent higher lycopene production. Substrates containing linoleic acid compounds led to high lycopene production. The data showed that the biosynthesis of lycopene starts in most cases simultaneously in the early growth phase even in trace amounts and the amount of lycopene formation increased continuously from 2 to 7 days.”
Two sentences have been added to abstract. One sentences has been revised as “The presence of lines 60-63: please make a separate last paragraph in the introduction section stating the objectives of the study.
The objectives of study has been separated from last paragraph.
line 83, section 2.3: a short paragraph should be included concerning the preparation of media no. 8 and 9 with orange peels. What was the procedure followed? Orange peels were used as solid particles in the basal medium? Were orange peels dried, grinded (particle size?)? Please give details.
In this study, the peel was used for powder. Therefor the drying of orange peel and preparation of orange peel powder and its size have been indicated on page 6 in section 2.3.
section 2.7, Analysis: Monitoring of carbon source could give results for better understanding of metabolism, so why the authors did not analyze the sugar consumption during fermentation? For instance, if you had the profile of sugars you could provide better discussion for the statement “Lycopene production continuously increased during the next 5 days. It reached maximum concentration after 7 days.” and see if there is any correlation between sugars consumption and lycopene production.
This was cited to a reference (22) on page 16.
Morphology was studied only in 4% glucose medium containing 1 % sunflower oil (as you mention in line 287)? Please make it clear in the materials and methods section. How biomass was monitored in the case of fermentations with orange peels? Usually fungus attach in solid particles. Clarify this with a few sentences.
The fermentation media were prepared using orange powder and the control was used without microbial inoculation. The control weight was subtracted from biomass obtained from media containing orange powder as indicated on page 9 at the end of last paragraph.
line 117: please write “High Performance Liquid Chromatography (HPLC)” and thereafter use only the abbreviation.
Long name has been indicated in the first writing place.
line 116, section 2.7.1: Was the sample automatically injected? The volume of injected sample was 20μm or 20 μl?
It was corrected as 20 μl.
lines 257-265: the paragraph must rewritten. There are small sentences without continuity, for example “Since oil slightly increases the 262 density of the fermentation media.”
Two sentences have removed from manuscript: “This would be due to retaining higher amount of oxygen in the media. Since oil slightly increases the density of the fermentation media and this density increased.” Since there is no any meaning to explain oxygen level.
line 261: correct “then” to “than”.
It was corrected
section 3.3 Morphological analysis: Do you have any images from morphological analysis to provide? It would add value to the manuscript. Explain why you present results from only one media.
In this study many substrates were used in the lycopene production. Therefore, it was impossible to study with all parameters. Morphological analysis was performed for only 4 % glucose medium containing 1 % sunflower oil that give higher amount of lycopene production.
Discussion section: What is the content of total phenolic content in lycopene? Do the authors validate in some way if there are significant quantities of other carotenoids? What is the yield of carotenoids in terms of the biomass?
Phenolic content of lycopene was not detected in research. The other carotenes were not studied in this research. The lycopene yield was not detected with respect to biomass and the yield of lycope with respect to biomass were not calculated. Therefore they were not indicated in this manuscript.
line 308: change “(6 %)” to “(6%)”
It was corrected through manuscript.
line 309: delete “the” from the sentence “cause the inhibition”
It was deleted.
line 329: give reference for the statement “…due to a higher amount of vitamin E and linoleic acid in sunflower oil.”
Two references add for this sentences on page 17.
lines 324-335: enrich your discussion with other publications. For example similar study with utilization of oils in the substrate has been demonstrated by Mantzouridou et al. 2006. Performance of Crude Olive Pomace Oil and Soybean Oil during Carotenoid Production by Blakeslea trispora in Submerged Fermentation. J. Agric. Food Chem., 2006, 54 (7), pp 2575–2581
It was enriched with more than one references (24) as indicated on page 17.
lines 344-350: enrich your discussion with other publications using renewable resources as substrates.
It was enriched with more than one references (24-29) as indicated on page 18.
line 357: change “12 V/min” to “12 V min-1”.
It was corrected as “12 L min-1”, since all others in L min-1.
Figures: error bars are missing from the figures. Please add them. Also, correct “Lycopen” to Lycopene” in the axis title and throughout the manuscript.
It was corrected throughout the manuscript. Figures bar have been revised.
Figure 1: please check the format of x-axis title for lycopene production, it should be “mg L-1”
“mg L-1” corrected throughout the manuscript.
Figure 8: change “Sunflavor oil” to “Sunflower oil” in the figure legend. Also, correct the figure caption since you mention that the medium contain both corn and sunflower oils. Also, be constant with units, for instance “3 L min-1” instead of “3 L/min”. Check for such mistakes all figure captions and the manuscript.
Figure captions has revised and 3 L min-1”has been corrected.
Other corrections on the bases of two reviewers
The manuscript English language were also checked by an English native speaker as indicated by second reviewer. According to the reviewers commends, 7 references have been added to Reference section and cited in the manuscript.
The manuscript was also revised and the following correction have been made on the manuscript:
1) Repeating sentence “Morphological forms of B. trispora zygospores are responsible for the production of the lycopene” has been removed from page 13 line 377.
2) The following repeating sentence has been removed from manuscript on page 3 line 94: “The orange powder was used in the preparation of fermentation medium. “
3) The following repeating sentence has been removed from manuscript on page 5 line 186: “The pH decreased and increased during the 7 days fermentation in the shake-flask and fermentor studies.”
4) The model and company for the following instruments have been added
- Degasser (on page 4, Section 2.71.) as “…….in a degasser (Bransonic 2200, 41 Eagle Road Danbury, Connecticut 06810-1961, USA).
- Grinder (on page 3 in first paragraph as “…………using a laboratory grinder (Roller type lab mill RM 1300, Erkaya, Ataşehir, İstanbul, Turkey), homogenized………”
Reviewer 2 Report
Considering the importance of natural pigments such as carotenoids in food and human health this study provides information on cheap and alternative ways to produce these chemical compounds. While fermentation is a known route for production of carotenes, use of cheap agricultural wastes such as orange peal is new and this study is novel in that regard. Having said this following are some of the points the authors have to consider reviewing and provide further clarity on their investigation. Only after that the manuscript can be considered for publication.
In fermentation media optimization usually Response Surface Methodology (RSM) is the usual design followed to determine the optimal level of substrate such as glucose, Sunflower oil, Orange peel etc used in the present study. Going my the present design the authors have used is not a correct way to develop any fermentation media optimization. So provide justification for this.
Why did authors choose sunflower and corn oil? The concentration etc?
What benchtop fermenter did they use? Provide detailed explanation of data collection and software used etc
Authors have used Lycopene cyclase inhibitor in their study, but its impact on lycopene production have not been discussed at all. Why? There is no table or figure about the impact of that inhibitor. Where is the control for this?
It is generally believed that secondary metabolites such as lycopene production increases under nutrient starvation situation or stress situation. That is one of the reason why 4% glucose yielded better lycopene than 6% while the biomass is better produced in 6% glucose
How did authors quantify biomass level in samples with orange peel?
How did authors confirmed the utilization of oils by fungi? Does this fungus have any transporters for oils? Please explain.
Author Response
Open Review
(x) I would not like to sign my review report
( ) I would like to sign my review report
English language and style
( ) Extensive editing of English language and style required
( ) Moderate English changes required
(x) English language and style are fine/minor spell check required
( ) I don't feel qualified to judge about the English language and style
Yes | Can be improved | Must be improved | Not applicable | |
Does the introduction provide sufficient background and include all relevant references? | (x) | ( ) | ( ) | ( ) |
Is the research design appropriate? | ( ) | (x) | ( ) | ( ) |
Are the methods adequately described? | ( ) | (x) | ( ) | ( ) |
Are the results clearly presented? | (x) | ( ) | ( ) | ( ) |
Are the conclusions supported by the results? | ( ) | (x) | ( ) | ( ) |
Comments and Suggestions for Authors
Considering the importance of natural pigments such as carotenoids in food and human health this study provides information on cheap and alternative ways to produce these chemical compounds. While fermentation is a known route for production of carotenes, use of cheap agricultural wastes such as orange peal is new and this study is novel in that regard. Having said this following are some of the points the authors have to consider reviewing and provide further clarity on their investigation. Only after that the manuscript can be considered for publication.
In fermentation media optimization usually Response Surface Methodology (RSM) is the usual design followed to determine the optimal level of substrate such as glucose, Sunflower oil, Orange peel etc used in the present study. Going my the present design the authors have used is not a correct way to develop any fermentation media optimization. So provide justification for this.
İnstead of Response Surface Methodology, we indicated possible critical parameters and their levels; these indications were studied. In other case, it was difficult to us for critical parameters and levels.
Why did authors choose sunflower and corn oil? The concentration etc?
We analyzed the literature results, these level was useful in the lycopene production, so they are used with our indicated levels.
What benchtop fermenter did they use? Provide detailed explanation of data collection and software used etc.
It was explained on page 6 in first paragraph.
The following repeating sentences were also removed from Section 2.5, second paragraph, since they indicated in Section 2.3.
“The compositions of base fermentation medium consisted of (g L-1): glucose 40, asparagine 2, yeast extract 1, KH2PO4 1.5 and MgSO4 0.5. Fermentation media for fermentor studies were prepared as given in Table 1.”
Authors have used Lycopene cyclase inhibitor in their study, but its impact on lycopene production have not been discussed at all. Why? There is no table or figure about the impact of that inhibitor. Where is the control for this?
In this research, without Lycopene cyclase inhibitor, the research was not performed. Therefore, there is no data to compare. There is no control without. Since it is the normal case to produce lycopene with the use of Lycopene cyclase inhibitor.
It is generally believed that secondary metabolites such as lycopene production increases under nutrient starvation situation or stress situation. That is one of the reasons why 4% glucose yielded better lycopene than 6% while the biomass is better produced in 6% glucose.
It was indicated on page as “% as “In this research, 4% glucose yielded better lycopene than 6% while the biomass is better produced in 6% glucose. Since, there is an indirect relationship between biomass and lycopene concentration.” On page 12 in first paragraph.
How did authors quantify biomass level in samples with orange peel?
The control medium for orange peel was added to manuscript on page 9 at the end of last paragraph as “The medium with %10 orange peel powder was used as a control; in biomass detection this was subtracted from biomass.”
How did authors confirmed the utilization of oils by fungi? Does this fungus have any transporters for oils? Please explain.
The utilization of oils by B. trispora has been indicated on page 17 as “Oils can be hydrolyzed to fatty acids and glycerol by mold exolipases [23].”
Submission Date
23 February 2019
Date of this review
03 Mar 2019 03:33:35

Round 2
Reviewer 1 Report
Figures have not revised according to reviewer's comments. The following must be corrected prior to any consideration for publication.
Figure 8: change “Sunflavor oil” to “Sunflower oil” in the figure legend.
"Lycopen" must rewritten to "Lycopene" in all figure titles.
Figures must have the same format, size, font size etc.
Author Response
Authors' Responses to Reviewer's Comments (Reviewer 1) (Review Report (Round 2) )
The following correction have been made on the manuscript as indicated by Reviewer 1:
1) Figure 8: change “Sunflavor oil” to “Sunflower oil” in the figure legend.
On Figure 8 “Sunflavor oil” has been correctes as “Sunflower oil”.
2) "Lycopen" must rewritten to "Lycopene" in all figure titles.
"Lycopen" has been corrected as "Lycopene" in all figure titles.
3) Figures must have the same format, size, font size etc.
Format, size and font of the figures have been revised and corrected. Now they have same format, size and font size.
